# A Universal Multi-Epitope Vaccine Design Against Porcine Reproductive and Respiratory Syndrome Virus via Bioinformatics and Immunoinformatics Approaches

**DOI:** 10.3390/vetsci11120659

**Published:** 2024-12-16

**Authors:** Xinnuo Lei, Zhi Wu, Qi Feng, Wenfeng Jia, Jun Xie, Qingkang Zhou, Jinzhao Ban, Shanyuan Zhu

**Affiliations:** 1Jiangsu Key Laboratory for High-Tech Research and Development of Veterinary Biopharmaceuticals, Engineering Technology Research Center for Modern Animal Science and Novel Veterinary Pharmaceutic Development, Jiangsu Agri-Animal Husbandry Vocational College, Taizhou 225300, China; xlei@jsahvc.edu.cn (X.L.); zhiwu@jsahvc.edu.cn (Z.W.); 2010010333@jsahvc.edu.cn (Q.F.); 2022010481@jsahvc.edu.cn (W.J.); xjjstz2018@163.com (J.X.); zhouqingkang1220@126.com (Q.Z.); 2020107040@stu.njau.edu.cn (J.B.); 2Ministry of Agriculture Key Laboratory of Animal Bacteriology, College of Veterinary Medicine, Nanjing Agricultural University, Nanjing 210095, China

**Keywords:** PRRSV, multi-epitope, universal vaccine, bioinformatics, immunoinformatics

## Abstract

Porcine reproductive and respiratory syndrome virus (PRRSV) is among the most harmful pathogens affecting the swine industry. Traditional inactivated or attenuated vaccines have proven inadequate for effective control, highlighting the need for novel, safe, and effective vaccine strategies. In this study, bioinformatics and immunoinformatics methods were used to rationally design a universal multi-epitope vaccine, which demonstrated excellent performance based on physicochemical properties, structural stability, immune receptor binding affinity, and immunogenicity. These attributes suggest that the designed vaccine holds significant promise as a novel, universal subunit vaccine candidate for preventing PRRSV infection.

## 1. Introduction

The swine industry constitutes a pivotal sector within the global agricultural economy, essential for food security and agricultural sustainability. However, its health and productivity are continually threatened by various pathogens that impact animal health, welfare, and the quality and safety of pork products. Porcine reproductive and respiratory syndrome virus (PRRSV), discovered nearly 40 years ago, remains one of the most challenging pathogens to control. With high variability, rapid transmission, and immunosuppressive characteristics, PRRSV causes reproductive disorders in sows and severe respiratory issues in piglets, presenting a critical concern worldwide [1,2].

PRRSV, a member of the *Arteriviridae* family and the *Betaarterivirus* genus, is divided into two species: *Betaarterivirus suid* 1 and *Betaarterivirus suid* 2 [3]. This enveloped, positive-sense single-stranded RNA virus has a ~15 kb genome encoding multiple open reading frames (ORFs). ORF1a and ORF1b encode large polyproteins pp1a and pp1ab, which are processed into at least 14 non-structural proteins (Nsps) essential for viral genome replication and transcription [4]. Additionally, PRRSV encodes eight structural proteins involved in virion assembly, infection, and immune evasion [5].

The high mutation rate of PRRSV has resulted in the emergence of numerous distinct variants, several of which serve as important reference strains in PRRSV research. For instance, the Lelystad virus (LV), the prototypical European PRRSV strain, was first identified in the Netherlands in 1991 [6]. VR-2332, a classical North American strain, was isolated in 1992 in the U.S. [7]. Early strains, such as CH-1a (identified in 1996), and highly pathogenic strains, like JXA1 (identified in 2006), which triggered outbreaks in China, serve as regional references [8]. More recently, NADC30 (identified in 2008) and NADC34 (identified in 2017) strains represent U.S. variants, with the highly pathogenic RFLP 1-4-4 lineage 1C emerging in 2020 and causing severe economic losses in the swine industry [9,10,11]. The sequences and pathogenic characteristics of these strains provide a foundation for studying PRRSV pathogenesis and advancing vaccine development.

Current commercial PRRSV vaccines, however, present notable limitations regarding safety and efficacy. Modified live vaccines (MLVs) offer partial protection but are associated with significant concerns, including virus shedding, reversion to virulence, recombination with wild-type variants, and limited cross-protection against heterologous variants [12]. Inactivated vaccines, while safer, suffer from inadequate immunogenicity [13]. Consequently, there is an urgent demand for novel vaccines that combine safety with high efficacy.

Epitope-based peptide vaccines offer a promising strategy by targeting specific pathogenic epitopes, eliminating toxic and allergenic viral components, and enabling precise immune responses against the pathogen [14]. They have been extensively utilized in the development of research targeting viral and bacterial diseases. For instance, Prakash et al. engineered a multi-epitope pan-variant SARS-CoV-2 vaccine comprising conserved human B and T cell epitopes from both structural and non-structural SARS proteins. This vaccine is capable of eliciting cross-protective immunity, facilitating viral clearance, and reducing the incidence, associated pulmonary pathology, and mortality [15]. Liu et al. employed reverse vaccinology and immunoinformatics to identify 11 conserved proteins in eight common serotypes of Streptococcus suis. They selected 16 candidate epitopes to design two promising epitope peptide vaccines, which demonstrated the potential to prevent Streptococcus suis infection in a murine model and exhibited excellent cross-protective effects [16]. The design and production of such vaccines are relatively straightforward, allowing for rapid adaptation to viral mutations and thus facilitating swift vaccine development [17]. Peptide vaccines recognized by B cells effectively enhance antibody production, while those targeting T cells stimulate a robust cellular immune response; together, these components enhance the overall immunogenicity of the vaccine [18,19].

Developing a safe and effective PRRSV vaccine remains a major challenge in veterinary medicine. While PRRSV epitope peptide vaccines show significant promise, research in this area is limited. A recent study employed immunoinformatics tools to develop a candidate vaccine incorporating B cell epitopes from the PRRSV GP3 and GP5 proteins, along with dominant protective cytotoxic T lymphocyte (CTL) and helper T lymphocyte (HTL) epitopes. However, this approach did not comprehensively address other PRRSV-encoded proteins or involve diverse variant sequences [20]. Our previous research demonstrated that linking reported PRRSV CTL epitopes with Fc molecules induced high cellular immunity in mice and prolonged the antigen half-life [3]. However, challenges remain in designing an epitope peptide vaccine that can simultaneously induce both humoral and cellular immunity and improve cross-protection against various PRRSV variants.

This study aimed to address these limitations by utilizing multiple bioinformatics tools to design a novel, universal anti-PRRSV multi-epitope peptide vaccine, optimizing it for efficacy, safety, and cross-protective potential. The design methodology in this study not only advances PRRSV vaccine development but also offers a new perspective for vaccine design against other viral diseases.

## 2. Materials and Methods

### 2.1. Acquisition and Alignment of Representative Strain Sequences

Sequences of seven representative PRRSV strains were obtained from the National Center for Biotechnology Information (NCBI) database (https://www.ncbi.nlm.nih.gov, accessed on 28 June 2024) with GenBank IDs as follows: M96262 (LV), U87392 (VR-2332), AY032626 (CH-1a), EF112445 (JXA1), MH500776 (NADC30), MF326985 (NADC34), and MW887655 (RFLP_1-4-4). Comparative analyses of genome sequences and protein identities among these strains, along with phylogenetic tree construction, were conducted using the Neighbor-Joining method in MEGA-11.

### 2.2. Prediction and Selection of Antigenic Epitopes

B cell neutralizing epitopes were identified based on published studies and verified across sequences from various strains. For Th epitope prediction, we employed the Immune Epitope Database (IEDB) MHC-II Binding Predictions tool (http://tools.iedb.org/mhcii/, accessed on 25 July 2024), using the NetMHCIIpan 4.1 BA method with a seven-allele HLA reference set and default length. Epitope-selection criteria were set at an IC50 ≤ 50 nM and %Rank ≤ 1. CTL Epitope prediction was conducted using NetMHCcons-1.1 (https://services.healthtech.dtu.dk/services/NetMHCcons-1.1/, accessed on 26 July 2024), with the following parameters: method = NetMHCcons; species = pig; alleles = SLA-1*0401, SLA-1*0801, SLA-2*0101, SLA-2*0401, SLA-2*0502, SLA-2*1001, and SLA-3*0401; peptide length = 9. Strong binders were defined as those with an IC50 ≤ 50 nM and %Rank ≤ 0.5. BepiPred 2.0 (http://tools.iedb.org/bcell/, accessed on 28 July 2024) was used to assess the antigenicity of different combinations of NE epitopes.

### 2.3. Evaluation of Physicochemical and Biochemical Properties

Expasy-ProtParam (https://web.expasy.org/protparam/, accessed on 30 July 2024) was used to estimate the physicochemical properties of the multi-epitope vaccine, while AllerTop v2.0 evaluated allergenicity, and Protein-Sol (https://protein-sol.manchester.ac.uk/, accessed on 30 July 2024) assessed protein solubility. VaxiJen v2.0 determined antigenicity, and Class I Immunogenicity assessed the immunogenic potential. AllerTop v2.0 (https://www.ddg-pharmfac.net/AllerTOP/method.html, accessed on 30 July 2024) and ToxiPred2 (https://webs.iiitd.edu.in/raghava/toxinpred2/batch.html, accessed on 30 July 2024) further analyzed potential allergenicity and toxicity.

### 2.4. Prediction, Refinement, and Validation of Protein Structure

The PSIPRED4.0 server (http://bioinf.cs.ucl.ac.uk/psipred, accessed on 29 July 2024) was used to predict the secondary structure of the multi-epitope vaccine. Tertiary structure modeling was performed using the AlphaFold3 server (https://alphafoldserver.com/, accessed on 10 August 2024). GalaxyRefine (https://galaxy.seoklab.org/cgi-bin/submit.cgi?type=REFINE, accessed on 15 August 2024) refined the 3D structure. Structural quality assessment was conducted using the SWISS-MODEL server (https://swissmodel.expasy.org/assess, accessed on 20 August 2024) and ProSA-web (https://prosa.services.came.sbg.ac.at/prosa.php, accessed on 20 August 2024).

### 2.5. Molecular Docking

Porcine TLR2 dimer, TLR4 dimer, SLA1, and SLA2 models were generated using AlphaFold3. Molecular docking was performed using the HDOCK server (http://hdock.phys.hust.edu.cn/, accessed on 23 August 2024), selecting the most stable model for further analysis. Docking results were visualized using PyMOL 3.1.

### 2.6. Immune Response Simulation

To evaluate immune responses, C-Immune Simulation (https://kraken.iac.rm.cnr.it/C-IMMSIM/, accessed on 10 September 2024) was used. Simulation parameters were optimized with 500 steps, administering immunizations on days 0, 28, and 70 without LPS. All other settings were set to default.

### 2.7. Codon Optimization and In Silico Cloning

Codon optimization for Escherichia coli and Spodoptera frugiperda was performed using ExpOptimizer (https://www.novopro.cn/tools/codon-optimization.html, accessed on 12 September 2024), translating the multi-epitope peptide sequence into a nucleotide sequence. The vaccine gene was cloned in silico into PET-28a(+) and pFastBac1 vectors using SnapGene v5.2.

## 3. Results

### 3.1. Selection of Reference Strains and Conserved Sequence Analysis

To guide vaccine design, seven historically significant strains of PRRSV were selected based on sequence data obtained from GenBank, in chronological order: LV, VR-2332, CH-1a, JXA1, NADC30, NADC34, and RFLP_1-4-4. A phylogenetic tree based on whole-genome sequences (Figure 1) revealed that these strains distinctly group into three major clusters, reflecting the three evolutionary phases of PRRSV. Additionally, the genomic identities of these seven strains, along with the amino acid sequence identities of their encoded proteins, are presented in Appendix A. This foundational information is critical for identifying consensus epitope sequences for further vaccine design.

### 3.2. Prediction and Selection of Epitopes

T-helper (Th) cell epitopes activate T cells, promoting cytokine secretion and enhancing immune responses. Using the IEDB MHC-II Binding Predictions tool, Th epitopes across the seven representative strains were predicted, applying a stringent threshold of IC50 ≤ 50 nM, percentile rank ≤ 1, and sequence conservation ≥5/7 strains. Seven peptides meeting these criteria were selected (Table 1), with additional details provided in Appendix A.

Cytotoxic T lymphocyte (CTL) epitopes, presented by MHC class I molecules, activate CD8+ T cells to trigger cellular immunity. Similar selection criteria were used for CTL epitopes, resulting in the initial identification of 65 CTL epitopes through IEDB MHC-I binding predictions. Of these, the 10 most promising CTL epitopes were selected (Table 2), including “RTAIGTPVY”, which has been previously reported [21].

For B cell epitopes, only neutralizing epitopes of PRRSV were retained, as non-neutralizing epitopes could potentially induce antibody-dependent enhancement (ADE). Known neutralizing epitopes include “SSHLQLIYNLT” in GP5 and “QAAAEVYEPGRS” in GP3. While neutralizing epitopes “WSFADGN” and “GVSAAQEKISFG” have been reported in GP4, sequence alignment revealed their presence solely in the LV strain (European genotype). Thirteen peptides were selected for further study based on their degree of conservation across strains (Table 3). The positions of all candidate epitopes within the protein amino acid sequences are indicated in Appendix A.

### 3.3. Construction of the Multi-Epitope Vaccine (THs-CTLs-NEs)

A multi-epitope vaccine construct, designated as THs-CTLs-NEs, was designed using immunoinformatics methods and includes 534 amino acids (Figure 2A). This vaccine comprises three main segments: NEs, THs, and CTLs. The TH segment consists of Th epitopes (Table 1), joined by “GPGPG” and “AAY” linkers. While the IEDB MHC-II binding predictions indicated a minimal impact of epitope order on binding affinity, linker adjustments were made to prevent the formation of unwanted epitopes with high binding affinity. Similarly, the CTL segment consists of the CTL epitopes (Table 2) joined in the same manner. NEs, which contain B cell epitopes, were linked by “GPGPG” and “EAAK” linkers. The sequence arrangement was refined using BepiPred 2.0, achieving 86% predicted B-cell epitope coverage in the NEs (Figure 2B). The C-terminal TT peptide was added to enhance immunogenicity [31]. To determine the optimal order of the three main segments, the 3D structures of six possible recombinant proteins were predicted using AlphaFold3 and assessed for structural integrity with the SWISS-MODEL Structure Assessment server, resulting in the final THs-CTLs-NEs arrangement (Appendix A).

Biochemical assessments indicated non-allergenicity as determined by AllerTop v.2.0, with Protein-Sol predicting a solubility probability of 0.293 in *Escherichia coli* (*E. coli*). VaxiJen v2.0 produced an antigenicity score of 0.4084, classifying the vaccine as an antigen. The Class I Immunogenicity assessment yielded a score of 4.43026, suggesting high immunogenicity. AllerTOP v2.0 and ToxiPred2 both identified the multi-epitope vaccine as non-allergenic and non-toxic. Based on these physicochemical and biochemical properties, the designed multi-epitope vaccine represents a promising candidate for further development.

### 3.4. Physicochemical and Biochemical Properties of the Multi-Epitope Vaccine

The physicochemical properties of the multi-epitope vaccine were estimated using Expasy-ProtParam, revealing a molecular weight of 57.11 kDa and a theoretical pI of 5.97. The instability index of 36.89 classifies the protein as stable, with a predicted half-life of more than 30 h in mammalian reticulocytes, over 20 h in yeast, and over 10 h in *E. coli*. Biochemical assessments indicated non-allergenicity as determined by AllerTop v.2.0, with Protein-Sol predicting a solubility probability of 0.293 in *E. coli*. VaxiJen v2.0 produced an antigenicity score of 0.4084, classifying the vaccine as an antigen. The Class I Immunogenicity assessment yielded a score of 4.43026, suggesting high immunogenicity. AllerTOP v2.0 and ToxiPred2 both identified the multi-epitope vaccine as non-allergenic and non-toxic. Based on these physicochemical and biochemical properties, the designed multi-epitope vaccine represents a promising candidate for further development.

### 3.5. Secondary and Tertiary Structure Prediction of the Multi-Epitope Vaccine

The secondary structure of the multi-epitope vaccine was predicted using the PSIPRED server, indicating the presence of coils (48.3%), helices (39.7%), and sheets (12.0%) among the structural residues (Figure 3A). Solubility assessments based on residue characteristics indicate proportions of small non-polar (45.7%), hydrophobic (16.5%), polar (25.1%), and aromatic plus cysteine residues (12.7%) (Figure 3B).

To confirm the proper presentation of epitopes and structural stability, tertiary structure simulation of the THs-CTLs-NEs construct was modeled using AlphaFold3, followed by refinement via the GalaxyREFINE web server (Figure 4A). Structural analysis confirmed that The THs, CTLs, and NEs segments predominantly form three distinct structural domains, each effectively displaying its respective epitopes. The SWISS-MODEL server was used for the structural assessment, and the resulting 3D model showed a MolProbity score of 0.97 and a Clash score of 0.88. The Ramachandran plot indicated that 96.8%, 0.56%, and 0.25% of residues fell within the Favored, outlier, and rotamer outlier regions, respectively (Figure 4B,C). The ProSA-web Z-score of −3.49 further validated the model’s quality (Figure 4D), indicating that the multi-epitope vaccine structure is stable.

### 3.6. Molecular Docking of the Constructed Vaccine with Porcine TLR2 and TLR4

Toll-like receptors (TLRs), particularly TLR2 and TLR4, are integral to immune system functions, facilitating pathogen recognition, immune response activation, and the modulation of immune cell behavior. To explore interactions between the multi-epitope vaccine and these receptors, tertiary structures of porcine TLR4 and TLR2 dimers were constructed using AlphaFold3. Molecular docking simulations were conducted using the HDOCK server to evaluate binding interactions between the multi-epitope vaccine and TLR4 or TLR2. Visualization analysis via PyMOL identified 60 amino acid residues in monomeric TLR2 and 63 in monomeric TLR4 that likely interact within a 5.0 Å range with 51 and 43 residues of THs-CTLs-NEs, respectively (Figure 5 and Appendix A). Notably, the interacting residues were predominantly localized within the THs and CTLs segments of the vaccine. These findings suggest that the designed multi-epitope vaccine demonstrates strong binding efficiency with TLR2 and TLR4.

### 3.7. In Silico Cloning of the Vaccine Candidate

In line with Escherichia coli codon preferences, the multi-epitope peptide sequence was reverse-translated into a nucleotide sequence using the ExpOptimizer server. This produced a nucleotide sequence with a codon adaptation index (CAI) of 0.81, GC content of 58.68%, and length of 1605 bp. The vaccine gene sequence was subsequently cloned into the PET-28a(+) vector at *BamHI* and *XhoI* restriction sites using SnapGene (Figure 6A). For potential eukaryotic expression, an alternative nucleotide sequence was generated based on Spodoptera frugiperda codon preferences, yielding a CAI of 0.84, GC content of 56.18%, and sequence length of 1605 bp. To aid in purification, a 6 × His tag was appended to the C-terminus. This eukaryotic construct was cloned into the pFastBac1 vector, also at *BamHI* and *XhoI* restriction sites (Figure 6B).

### 3.8. Immunogenicity Simulation of the Epitope-Based Vaccine

The ability of the vaccine to induce both humoral and cellular immune responses is crucial in combating viral infections. Using the C-ImmSim server, we evaluated the primary and secondary immune responses of the host following administration of the vaccine candidate. After optimizing the immunization schedule, high levels of B cells, Th cells, and cytotoxic T cell (TC) were observed, indicating robust immune responsiveness. Figure 7A illustrates the antibody levels induced after three immunizations, with peak antibody levels reached at around day 80 post-immunization. Specific IgG levels, dominated by IgG1, reached a titer of approximately 1:100,000, while IgM titers peaked at around 1:1,125,000, with a combined IgM + IgG titer of about 1:225,000. Additionally, interferon-gamma (IFN-γ) levels rose significantly, reaching approximately 420,000 ng/mL after the first immunization. IL-2 peaked at around 700,000 ng/mL after the second immunization (Figure 7B).

Total and active B cell populations showed marked increases, peaking at approximately 780 cells/mm^3^ and 770 cells/mm^3^, respectively, after the third immunization (Figure 7C,D). Total Th cells and activated Th cell populations also increased significantly, reaching nearly 11,700 cells/mm^3^ and 7600 cells/mm^3^ after the second immunization (Figure 7E,F). Additionally, the TC population rapidly increased to 1150 cells/mm^3^ post-initial immunization (Figure 7G) and remained at between 600 and 1000 cells/mm^3^ up to day 165 (Figure 7H). Collectively, these simulations suggest that the developed vaccine successfully induces a potent immune response in mammals following three doses.

## 4. Discussion

This study introduces a novel, universal multi-epitope vaccine, THs-CTLs-NEs, targeting PRRSV. Using advanced bioinformatics and immunoinformatics tools, this vaccine was designed to be non-toxic and non-allergenic, with excellent antigenicity and immunogenicity. Additionally, the vaccine demonstrated strong potential for interactions with TLR2 and TLR4 receptors, effectively stimulating both humoral and cellular immune responses. This research provides a theoretically feasible approach for the development of safe and effective PRRSV subunit vaccines, addressing the limitations of current PRRSV vaccines, such as limited protection, insufficient cross-protection, and biosafety risks. Furthermore, it offers valuable insights into the design of vaccines for other viral diseases.

Research on peptide vaccines based on PRRSV epitopes remains limited. Our previous work focused on designing multi-epitope vaccines based on a single strain, including CTL epitopes, but these lacked the ability to effectively induce humoral immunity and cross-protection [3]. Considering the high variability of PRRSV, the goal of vaccine research is to develop a universal vaccine capable of eliciting robust humoral and cellular immune responses to overcome the limitations of current commercial vaccines. In this study, we incorporated previously reported neutralizing epitopes along with corresponding epitopes from seven representative PRRSV strains, effectively reducing the risk of antibody-dependent enhancement (ADE) and enhancing broad applicability. Regarding T cell epitopes, we used epitope prediction software to analyze all coding proteins, including structural and non-structural proteins, of seven representative PRRSV strains, selecting 7 Th epitopes and 10 CTL epitopes under strict criteria. Among these, only 1 CTL epitope had been previously reported, whereas 22 CTL epitopes identified in earlier studies passed preliminary screening but failed to meet precise selection criteria [3]. Thus, the epitopes selected in this study theoretically exhibit stronger binding affinity for SLA-1 or SLA-2, indicating higher potential for practical applications. A recent study aligns with our epitope-selection strategy; it developed a multi-epitope vaccine against SARS-CoV-2, incorporating conserved B and T cell epitopes derived from both structural and non-structural viral proteins. This vaccine demonstrated the potential to induce cross-protective immunity, enhance viral elimination, and decrease infection rates [15]. Although our study does not present in vivo immunization results for PRRSV peptide vaccines, the vaccine design approach is at least theoretically sound.

The design principles of vaccines are generally based on the characteristics of prevalent viral strains. Many studies and commercialized vaccines rely on a single prevalent strain for their development. However, this approach most likely results in limited cross-protective efficacy. The novelty of our study in vaccine design lies in the selection of epitopes based on the sequences of seven representative PRRSV strains. These strains represent the most typical strains from past and current epidemics and have been extensively studied in the literature. The epitopes identified from these sequences, particularly those with high conservation, cover the corresponding epitopes of the vast majority of known PRRSV strains. Therefore, vaccines designed based on these epitopes theoretically possess significant broad protective potential. While the emergence of novel PRRSV variants in the future may pose challenges to this design, the methodology employed here can be refined and applied to future iterations of PRRSV vaccines.

In similar studies on epitope-based vaccines, the arrangement of epitopes is often achieved by linking them with commonly used linkers, such as “GPGPG”, “EAAK”, and “AAY” [31,32,33]. However, these linkers can unintentionally create non-target epitopes with high binding affinity, a frequently overlooked issue in the design of epitope-based vaccines. To address this problem, this study employed epitope-prediction software to evaluate various linkage strategies, ensuring optimal binding affinity. Additionally, we utilized a comprehensive suite of bioinformatics tools to conduct multidimensional predictions and assessments of the vaccine’s secondary and tertiary structures. The TH, CTL, and NE segments formed relatively independent structural domains within the vaccine and demonstrated strong binding affinities for TLR2 and TLR4 receptors, particularly within the TH and CTL segments, as observed in subsequent molecular docking studies. These findings substantiate the theoretical robustness and efficacy of the multi-epitope vaccine design. Immune simulation results further revealed that the multi-epitope vaccine developed in this study could elicit strong humoral and cellular immune responses. Notably, significant enhancements were observed in IgG, IFN-γ, IL-2, B cell populations, Th cell populations, and Tc cell populations. These indicators surpassed the results reported in similar studies [32,34,35] and maintained a strong immune response over an extended period.

Although the multi-epitope vaccine designed in this study demonstrates remarkable immunogenicity against PRRSV in silico, several challenges may arise during its production and practical applications. First, while our predicted results support the vaccine’s potential, they do not conclusively reflect actual immunogenicity, particularly with respect to protective efficacy and cross-protection. Therefore, experimental validation is essential before clinical applications. Second, although epitope-based peptide vaccines offer advantages, such as an efficient design and high safety, they also face challenges, including a short half-life and limited delivery efficiency [14]. Potential solutions to these issues include employing sustained-release adjuvants or Fc molecules to prolong in vivo stimulation, incorporating targeting peptides or molecules to enhance delivery efficiency, or utilizing live viral vector systems [3,36,37,38]. Furthermore, the introduction of biological adjuvants may enhance the immunogenicity of epitope-based vaccines. 

Nevertheless, it is undeniable that the reliability of bioinformatics and immunoinformatics tools has been validated in numerous studies, and their application in vaccine design, particularly in protein structure modeling, receptor docking, and immunoinformatics approaches, is transforming the field of vaccine development. Notable advances have been achieved in diseases, such as trypanosomiasis, tuberculosis, and SARS-CoV-2, using these methods [39,40,41]. These tools enhance vaccine stability and specificity, streamline the screening process for protective antigens from genomic data, and improve vaccine safety and efficacy predictions, making vaccine development more efficient and cost-effective. The application of bioinformatics provides essential guidance in selecting vaccine candidates and optimizing production, suggesting that future vaccine designs may become increasingly precise and efficient.

## Figures and Tables

**Figure 1 vetsci-11-00659-f001:**
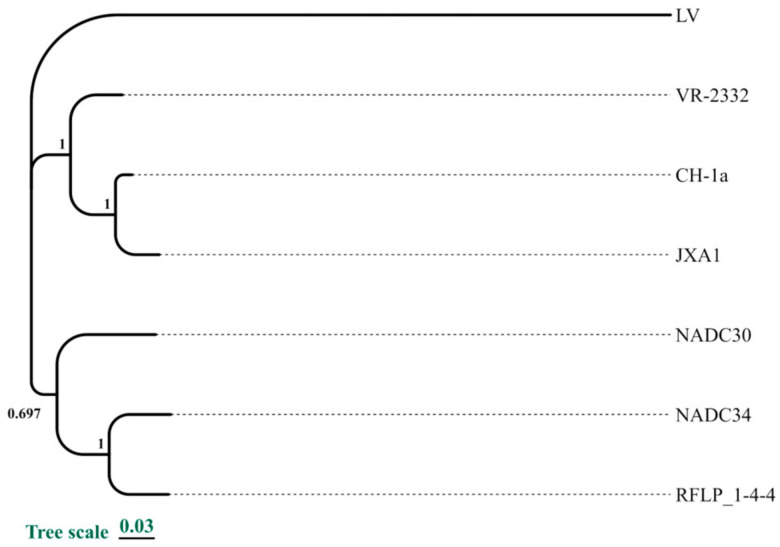
Phylogenetic analysis of seven representative PRRSV strains. A Neighbor-Joining (NJ) tree was constructed based on full-length genomic sequences using the p-distance model with 1000 bootstrap replicates to assess the phylogenetic relationships among these strains.

**Figure 2 vetsci-11-00659-f002:**
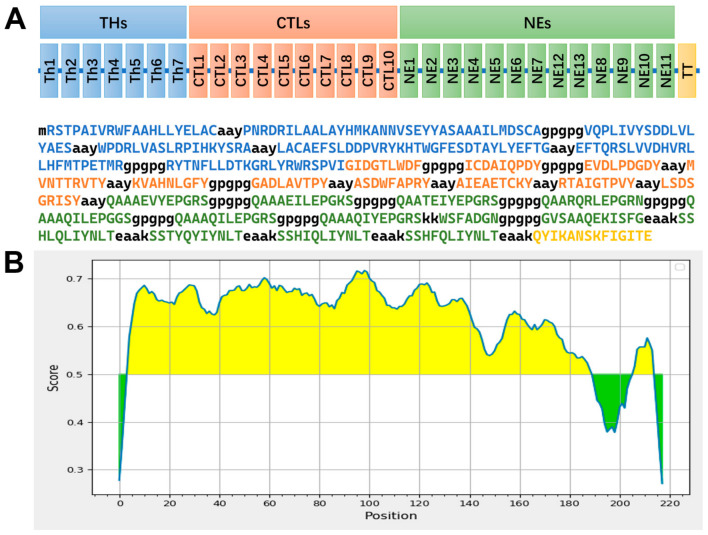
Design of the finalized multi-epitope vaccine THs-CTLs-NEs. (**A**) Epitope concatenation order and sequence information. (**B**) BepiPred 2.0 assessment of the final NEs, with yellow regions indicating predicted B cell epitopes, while green regions are likely not to correspond to B cell epitopes.

**Figure 3 vetsci-11-00659-f003:**
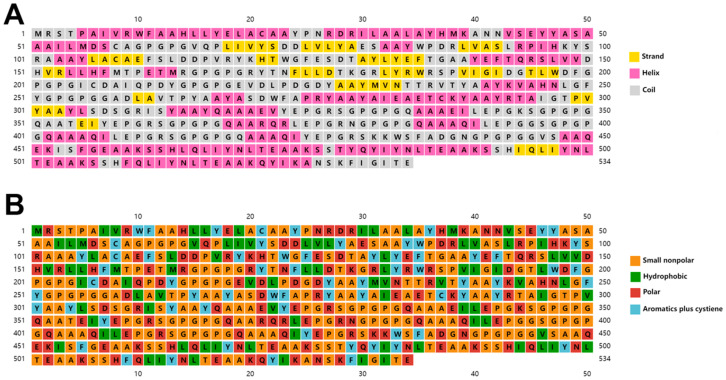
Predicted secondary structure of the vaccine construct. (**A**) Distribution of α-helices, β-strands, and coils among the amino acid residues. (**B**) Distribution of small non-polar, hydrophobic, polar, and aromatic residues.

**Figure 4 vetsci-11-00659-f004:**
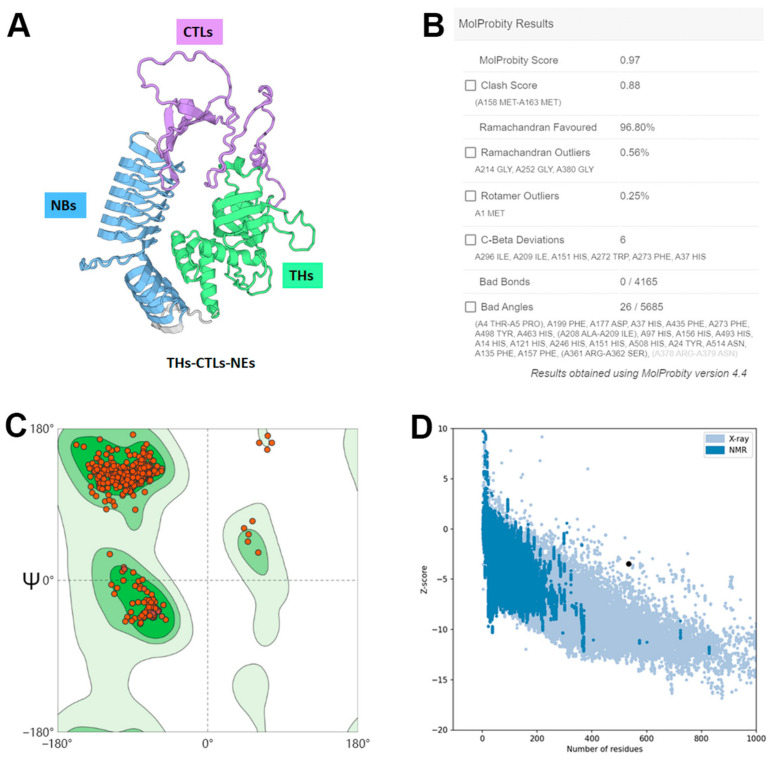
Evaluation of the tertiary structure of THs-CTLs-NEs. (**A**) AlphaFold3-predicted tertiary structure, optimized by GalaxyREFINE. (**B**) Structural assessment parameters, focusing on Ramachandran Favored regions. (**C**) Ramachandran plot with red dots representing residues in Favored, outlier, and rotamer outlier regions. A higher concentration in dark regions indicates better structural quality. (**D**) ProSA-web model quality assessment, where the Z-score evaluates the sample within the typical range for proteins of a similar size.

**Figure 5 vetsci-11-00659-f005:**
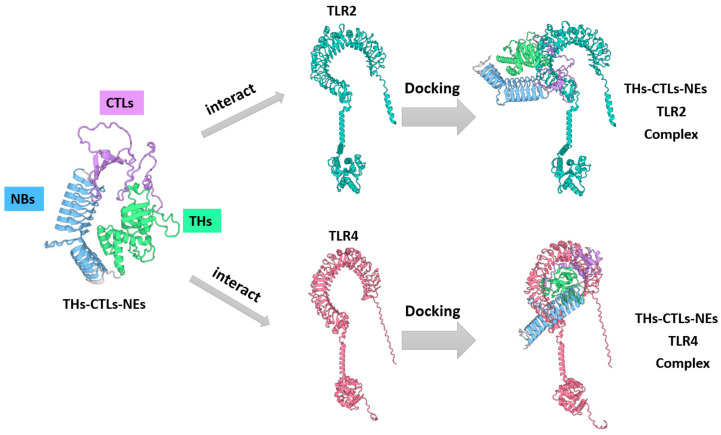
Molecular docking of THs-CTLs-NEs with porcine TLR2 and TLR4 via the HDOCK server.

**Figure 6 vetsci-11-00659-f006:**
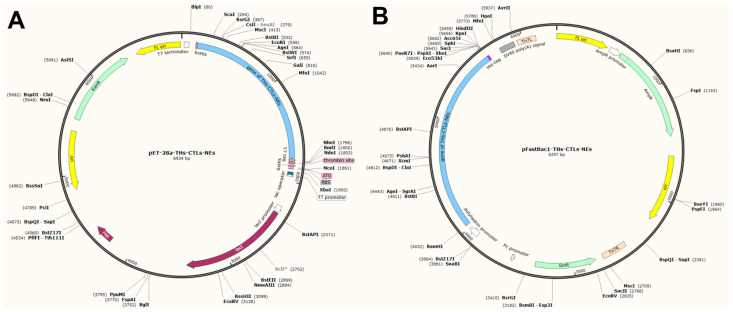
In silico cloning of the vaccine candidate into PET-28a(+) for prokaryotic expression (**A**) and pFastBac1 for eukaryotic expression (**B**).

**Figure 7 vetsci-11-00659-f007:**
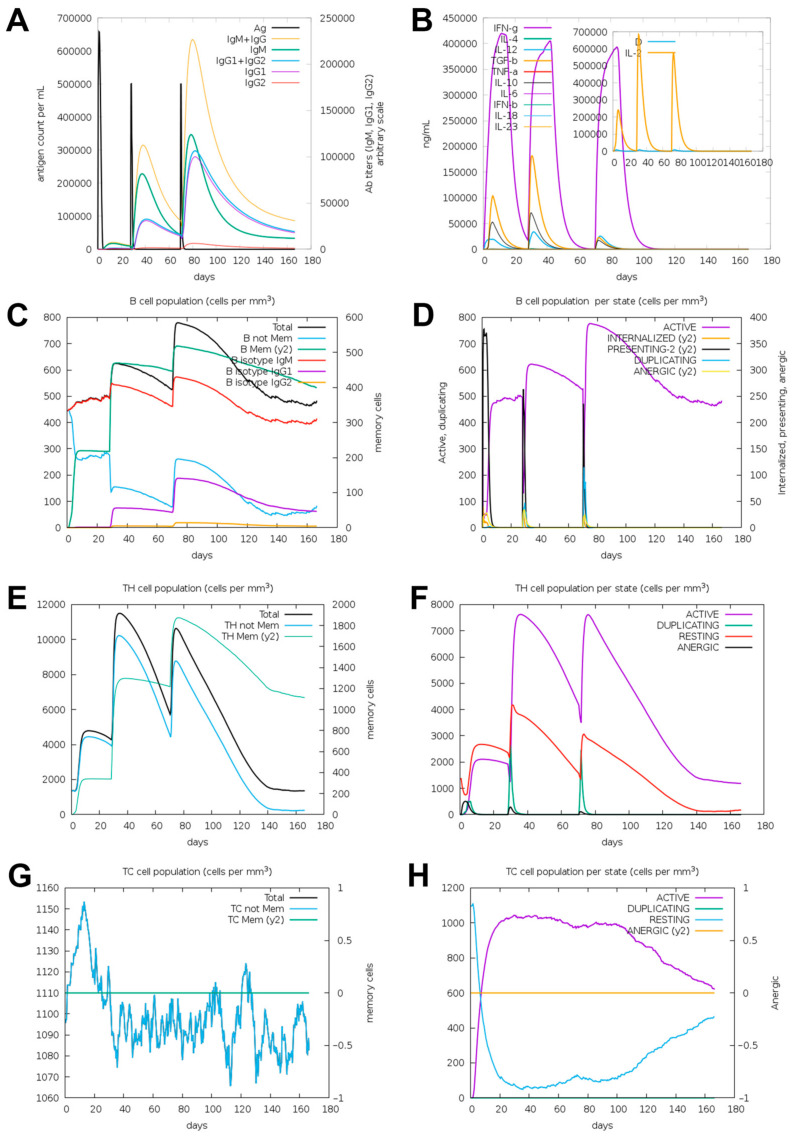
Immune response simulation of THs-CTLs-NEs via the C-ImmSim server across three injections: (**A**) antibody titers, (**B**) cytokine levels, (**C**) total B cell population, (**D**) active B cell population, (**E**) total Th cell population, (**F**) active Th cell population, (**G**) total TC cell population, and (**H**) active TC cell population.

**Table 1 vetsci-11-00659-t001:** Candidate Th epitopes.

Epitope	Protein	Sequence	Length	Core Sequence	Epitope Conservation	High Binders
Th1	Nsp9	RSTPAIVRWFAAHLLYELAC (H/N)	20	VRWFAAHLL	6/7	12
Th2	Nsp9	PNRDRILAALAYHMKANNVSEYYASAAAILMDSCA (N/S)	35	ILAALAYHM YYASAAAIL	6/7	28
Th3	Nsp9	VQPLIVYSDDLVLYAES	17	VYSDDLVLY	6/7	9
Th4	Nsp11	WPDRLVASLRPIHKYSRA (A/T)	18	VTSLRPIHK	6/7	12
Th5	Nsp12	LACAEFSLDDPVRYKHTWGFESDTAYLYEFTG (R/K)	32	FESDTAYLY FSLDDPVRY	6/7	22
Th6	GP4	EFTQRSLVVDHVRLLHFMTPETMR	24	LVVDHVRLL LLHFMTPET	5/7	9
Th7	GP5	RYTNFLLDTKGRLYRWRSPVI (R/K)	21	FLLDTKGRL	5/7	8

Note: some predicted Th epitope sequences, each 15 amino acids in length, overlap and have been consolidated into single epitope sequences. Amino acids highlighted in red indicate strain-specific mutations. The count of high-binding peptides indicates the number of integrated sequences, with higher counts suggesting increased MHC II recognition potential.

**Table 2 vetsci-11-00659-t002:** Candidate CTL epitopes.

Epitope	Protein	Sequence	Length	Epitope Conservation
CTL1	nsp9	GIDGTLWDF	9	5/7
CTL2	nsp10	ICDAIQPDY	9	6/7
CTL3	nsp10	EVDLPDGDY	9	5/7
CTL4	nsp10	MVNTTRVTY	9	5/7
CTL5	nsp11	KVAHNLGFY	9	6/7
CTL6	nsp12	GADLAVTPY	9	6/7
CTL7	GP2a	ASDWFAPRY	9	6/7
CTL8	GP2a	AIEAETCKY	9	6/7
CTL9	GP4	**RTAIGTPVY**	9	6/7
CTL10	N	LSDSGRISY	9	6/7

**Table 3 vetsci-11-00659-t003:** Candidate neutralizing epitopes.

Epitope	Protein	Sequence	Length	Strain	References
NE1	GP3	QAAAEVYEPGRS	12	CH-1a	[22,23]
NE2	GP3	QAAAEILEPGKS	12	JXA1	N/A
NE3	GP3	QAATEIYEPGRS	12	VR-2332	N/A
NE4	GP3	QAARQRLEPGRN	12	LV	N/A
NE5	GP3	QAAAQILEPGGS	12	NADC30	N/A
NE6	GP3	QAAAQILEPGRS	12	RFLP_1-4-4	N/A
NE7	GP3	QAAAQIYEPGRS	12	NADC34	N/A
NE8	GP5	SSHLQLIYNLT	11	VR-2332 RFLP_1-4-4 NADC30 NADC34	[24,25]
NE9	GP5	SSTYQYIYNLT	11	LV	[26]
NE10	GP5	SSHIQLIYNLT	11	JXA1	N/A
NE11	GP5	SSHFQLIYNLT	11	CH-1a	N/A
NE12	GP5	WSFADGN	7	LV	[27]
NE13	GP4	GVSAAQEKISFG	12	LV	[28,29,30]

## Data Availability

The data presented in this study are available within the article.

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
