# Peer review of "A Universal Multi-Epitope Vaccine Design Against Porcine Reproductive and Respiratory Syndrome Virus via Bioinformatics and Immunoinformatics Approaches"

_vetsci, 2024, doi:10.3390/vetsci11120659_

Round 1

Reviewer 1 Report

Comments and Suggestions for Authors

The manuscript presents an interesting study using bioinformatics to design and evaluate a vaccine against PRRSV. This strategy has been used for other pathogens, not for pathogens affecting pigs, at least to my knowledge. The manuscript is easy to read, and the methods and results are straightforward. The introduction requires adding other papers that have used peptides as vaccine candidates. Also, the discussion section must use those papers to discuss the limitations and advantages of this new "protein" made by peptides. Notably, lines 336-343 must be deleted or rewritten, the aim stated in these lines is not supported. The bioinformatic analysis is an interesting approach to the possible immunogenicity of this hypothetical protein. However, it must be clarified if the protein will be expressed satisfactorily. More importantly, there is no support to suggest that this protein will induce broad cross-protection against PRRS. The authors must avoid these kinds of suppositions that could distract the readers' attention. 

Author Response

Comments 1: The manuscript presents an interesting study using bioinformatics to design and evaluate a vaccine against PRRSV. This strategy has been used for other pathogens, not for pathogens affecting pigs, at least to my knowledge. The manuscript is easy to read, and the methods and results are straightforward.

Response 1: We sincerely thank you for your thoughtful comments and valuable suggestions. We have addressed the points raised as follows:

Comments 2: The introduction requires adding other papers that have used peptides as vaccine candidates.

Response 2: We have incorporated two relevant studies (lines 77-87) that highlight the effectiveness of epitope-based vaccines. These references provide context for our approach and support the use of peptides as vaccine candidates.

Comments 3: Also, the discussion section must use those papers to discuss the limitations and advantages of this new "protein" made by peptides.

Response 3: We have revised the discussion section (lines 398-414) to better address the advantages and limitations of peptide-based vaccines. We now discuss both the potential and challenges of using such vaccines, acknowledging their immunogenicity and effectiveness while also considering their limitations, such as the need for experimental validation and issues with peptide stability.

Comments 4: Notably, lines 336-343 must be deleted or rewritten, the aim stated in these lines is not supported. The bioinformatic analysis is an interesting approach to the possible immunogenicity of this hypothetical protein. However, it must be clarified if the protein will be expressed satisfactorily. More importantly, there is no support to suggest that this protein will induce broad cross-protection against PRRS. The authors must avoid these kinds of suppositions that could distract the readers' attention.

Response 4: We have carefully reviewed and revised the content in lines 336-343 to remove speculative statements that may have caused confusion. Specifically, we have clarified that the bioinformatics analysis provides valuable insights into the potential immunogenicity of the designed peptides, but we have avoided overstatements regarding the vaccine’s ability to induce broad cross-protection against PRRSV without experimental validation. We agree with the reviewer that these claims needed further refinement and have focused on presenting our findings in a more evidence-based manner. The rewritten content can be found in lines 368-379 of the revised manuscript.

We believe that these revisions have significantly improved the clarity and accuracy of the manuscript. We are grateful for your constructive feedback, which has helped us refine our work.

Reviewer 2 Report

Comments and Suggestions for Authors

Dear authors,

Congratulations. I think the work is good, explicit and worthy.

I think that the point 5 "Significance of the Study" should be the first point of the discussion. I think the readjustment makes the importance of the study clearer.

However, I think that the limitations of the study and what problems the authors predict will exist when the vaccine is produced and applied in farms' management must also be clarified.

Author Response

Comments 1: Dear authors, congratulations. I think the work is good, explicit and worthy.

Response 1: We sincerely thank you for your positive evaluation and thoughtful suggestions. In response to your comments, we have made the following revisions:

Comments 2: I think that the point 5 "Significance of the Study" should be the first point of the discussion. I think the readjustment makes the importance of the study clearer.

Response 2: As per your suggestion, we have moved the section "Significance of the Study" to the beginning of the discussion. We then emphasized the innovative aspects of our vaccine design, particularly the selection of epitopes derived from seven representative PRRSV variants, which enhances the vaccine's potential for broad protection.

Comments 3: However, I think that the limitations of the study and what problems the authors predict will exist when the vaccine is produced and applied in farms' management must also be clarified.

Response 3: We have now provided a more detailed discussion of the limitations of our study and the potential challenges in translating our vaccine design into a practical farm application. Specifically, we highlighted issues such as the need for experimental validation of the vaccine's immunogenicity, as well as concerns related to the short half-life and delivery efficiency of peptide-based vaccines. To address these challenges, we proposed several possible solutions, including optimizing peptide stability and exploring advanced delivery systems.

In addition to discussing the limitations, we have expanded on the practical considerations for implementing the vaccine in farm management. This includes addressing factors such as cost, scalability, and the feasibility of large-scale vaccine production and administration in farm settings.

We believe these revisions strengthen the manuscript by providing a comprehensive view of both the contributions and challenges of our study. We appreciate your valuable feedback, which has greatly improved the clarity and depth of our discussion.

Reviewer 3 Report

Comments and Suggestions for Authors

The manuscript describes the design of a universal multiple-epitope vaccine against one of the most important diseases that affects pigs, the porcine reproductive and respiratory virus, using bioinformatic tools. Although, I found the manuscript an enjoyable read, the methodology has several limitations. The bioinformatic design is based on seven selected strains. Taking into account that the main challenge of this virus is the genetic variability due to its mutation and recombination, is imprecise to obtain a vaccine candidate among them. Thus, I recommend to resubmit the manuscript after a major revision of the methodology.

Author Response

Comments 1: The manuscript describes the design of a universal multiple-epitope vaccine against one of the most important diseases that affects pigs, the porcine reproductive and respiratory virus, using bioinformatic tools. Although, I found the manuscript an enjoyable read, the methodology has several limitations.

Response 1: We sincerely thank you for your feedback on our manuscript, and we would like to address the comment on the limitations of the methodology.

Comments 2: The bioinformatic design is based on seven selected variants. Taking into account that the main challenge of this virus is the genetic variability due to its mutation and recombination, is imprecise to obtain a vaccine candidate among them. Thus, I recommend to resubmit the manuscript after a major revision of the methodology.

Response 2:

On the Selection of Seven Representative Variants: We fully acknowledge the challenge posed by the high genetic variability of PRRSV, which can result from mutation and recombination. We agree that relying on a limited number of variants may not fully capture the breadth of genetic diversity in the virus. However, our selection of seven representative variants was based on their prevalence, historical significance, and the large body of research focused on these variants. These variants are well-characterized and have been widely studied in both research and commercial vaccine contexts.

While we recognize the potential for new variants to emerge and challenge this approach, the epitopes we selected based on these variants, particularly those epitopes with high sequence conservation, are expected to cover the majority of known PRRSV variants, as indicated by multiple sequence alignments (data not shown). This design approach allows for a broader coverage of the virus and is in line with vaccine design strategies that focus on prevalent or circulating variants to achieve optimal protection.

On the Theoretical Nature of the Design and Future Adaptation: We acknowledge that bioinformatic vaccine design is inherently theoretical at this stage, but it serves as a valuable starting point for experimental validation. Similar to other predictive tools like AlphaFold and molecular docking software, bioinformatics provides a framework for guiding experimental efforts. While future variants of PRRSV may challenge our current design, we believe that the methodology used in this study provides a solid foundation for future iterations of PRRSV vaccines. Furthermore, this design can be adapted as new data on circulating variants and emerging variants become available.

On Experimental Validation: We are aware of the need for experimental validation to assess the true efficacy and reliability of the vaccine. To this end, we have already initiated the synthesis and expression of the designed vaccine sequences and plan to conduct animal immunization experiments in the next phase of our work. These experiments will provide a more accurate assessment of the vaccine's immunogenicity and potential for cross-protection.

Revised Discussion Section: To address your concerns and improve the clarity of our manuscript, we have rewritten the discussion section to present the methodology in a more balanced and objective manner. We have emphasized the rationale behind selecting representative variants and highlighted the potential for future improvements to our design strategy.

We believe that these revisions and clarifications address your concerns while reinforcing the value of our approach. We hope that our responses meet your expectations and that the revised manuscript is now suitable for publication.

Round 2

Reviewer 1 Report

Comments and Suggestions for Authors

Line 398. "...though the multi-epitope vaccine designed in this study demonstrates protective potential against PRRSV in silico," The statement is not supported and must be removed.

Line 404. "ease of production" Remove. You do not have information to support it.

Line 405-408. This review provides additional options to increase protein delivery efficiency. DOI: 10.3389/fimmu.2023.1080238

Line 410-414. "Additionally, in the context of farm management...." Irrelevant, remove it.

Comments on the Quality of English Language

No problems

Author Response

Comments 1: Line 398. "...though the multi-epitope vaccine designed in this study demonstrates protective potential against PRRSV in silico," The statement is not supported and must be removed.

Response 1: We agree that the statement about the "protective potential" of the multi-epitope vaccine against PRRSV in silico is premature and lacks sufficient supporting data. Our study only evaluated the immunogenicity of the vaccine. Therefore, we have revised the statement to“Although the multi-epitope vaccine designed in this study demonstrates remarkable immunogenicity against PRRSV in silico.”We believe this modification is more precise while maintaining the integrity of the paragraph.

Comments 2: Line 404. "ease of production" Remove. You do not have information to support it.

Response 2: Thank you for pointing out this unsupported claim. We have removed this phrase from the manuscript to ensure scientific rigor.

Comments 3: Line 405-408. This review provides additional options to increase protein delivery efficiency. DOI: 10.3389/fimmu.2023.1080238

Response 3: We have reviewed the recommended article (DOI: 10.3389/fimmu.2023.1080238) and found it highly relevant to our discussion. The reference has been included, and the discussion has been expanded accordingly.

Comments 4: Line 410-414. "Additionally, in the context of farm management...." Irrelevant, remove it.

Response 4: We acknowledge that this section is not directly related to the core focus of our study. To improve the relevance and coherence of the manuscript, this portion has been removed.

We deeply appreciate your constructive critique and suggestions, which have significantly enhanced the quality and clarity of our manuscript.